# Insight into the Crosstalk between Photodynamic Therapy and Immunotherapy in Breast Cancer

**DOI:** 10.3390/cancers15051532

**Published:** 2023-02-28

**Authors:** Hongzhong Jin, Shichong Liao, Feng Yao, Juanjuan Li, Zhiliang Xu, Kailiang Zhao, Ximing Xu, Shengrong Sun

**Affiliations:** 1Department of Breast and Thyroid Surgery, Renmin Hospital of Wuhan University, 238 Jiefang Road, Wuhan 430060, China; 2Central Laboratory, Renmin Hospital of Wuhan University, 238 Jiefang Road, Wuhan 430060, China; 3Department of Hepatobiliary Surgery, Renmin Hospital of Wuhan University, 238 Jiefang Road, Wuhan 430060, China; 4Cancer Center, Renmin Hospital of Wuhan University, 238 Jiefang Road, Wuhan 430060, China

**Keywords:** breast cancer, T cell, macrophage, photodynamic therapy, ROS

## Abstract

**Simple Summary:**

Immunotherapy has made tremendous clinical progress in breast cancer. However, in some patients, the response rate to immunotherapy is low because the tumor microenvironment (TME) is highly immunosuppressive and the tumors are not sufficiently immunogenic. Photodynamic therapy (PDT) can not only kill tumor cells directly but also induce immunogenic cell death (ICD), which provides antitumor immunity. This review discusses the recent advances in crosstalk between photodynamic therapy and immunotherapy in breast cancer, aiming to provide new perspectives on the treatment of breast cancer.

**Abstract:**

Breast cancer (BC) is the world’s second most frequent malignancy and the leading cause of mortality among women. All in situ or invasive breast cancer derives from terminal tubulobular units; when the tumor is present only in the ducts or lobules in situ, it is called ductal carcinoma in situ (DCIS)/lobular carcinoma in situ (LCIS). The biggest risk factors are age, mutations in breast cancer genes 1 or 2 (BRCA1 or BRCA2), and dense breast tissue. Current treatments are associated with various side effects, recurrence, and poor quality of life. The critical role of the immune system in breast cancer progression/regression should always be considered. Several immunotherapy techniques for BC have been studied, including tumor-targeted antibodies (bispecific antibodies), adoptive T cell therapy, vaccinations, and immune checkpoint inhibition with anti-PD-1 antibodies. In the last decade, significant breakthroughs have been made in breast cancer immunotherapy. This advancement was principally prompted by cancer cells’ escape of immune regulation and the tumor’s subsequent resistance to traditional therapy. Photodynamic therapy (PDT) has shown potential as a cancer treatment. It is less intrusive, more focused, and less damaging to normal cells and tissues. It entails the employment of a photosensitizer (PS) and a specific wavelength of light to create reactive oxygen species. Recently, an increasing number of studies have shown that PDT combined with immunotherapy improves the effect of tumor drugs and reduces tumor immune escape, improving the prognosis of breast cancer patients. Therefore, we objectively evaluate strategies for their limitations and benefits, which are critical to improving outcomes for breast cancer patients. In conclusion, we offer many avenues for further study on tailored immunotherapy, such as oxygen-enhanced PDT and nanoparticles.

## 1. Introduction

Breast cancer (BC), the second most common malignancy, accounts for 16.1% of all new cancer cases in women [1]. Survival rates for BC vary widely around the world. In developed countries, the estimated 5-year survival rate is 80%; in developing countries, it is less than 40%. According to the WHO classification 2019, breast cancer is classified into four types based on molecular and histologic findings: luminal A-like, luminal B-like, HER2-positive, and basal-like (triple negative) [2,3]. Although advances in early detection and therapy have led to a 38% decrease in BC mortality, cancer metastasis and resistance to therapy are significant barriers to the successful treatment of BC. Immunotherapy, which stimulates the host immune system to induce anticancer immune responses, has opened a new chapter in the treatment of malignant tumors in recent years [4]. In BC, increasing scientific evidence supports that cancers cause local immune dysregulation by suppressing the innate immune system, tumor-induced inflammation, and suppressing the adaptive T and B cell immune response in situ [5]. Despite the positive outlook, immunotherapy helps only some cancer patients, and the lack of tumor specificity results in particular immunotoxicity in a significant number of treated individuals. An increasing number of researchers are investigating new nanomedicines for PDT-assisted BC immunotherapy.

Photodynamic therapy (PDT) is a cutting-edge, noninvasive therapy with intriguing therapeutic applications in cancer treatment. It consists of three major parts: PS, visible light with a particular wavelength, and molecular oxygen [6,7]. PS molecules absorb appropriate wavelength light and begin activation mechanisms that result in the selective death of inappropriate cells. PS act as catalysts when they absorb visible light and then convert molecular oxygen into a series of highly reactive oxygen species (ROS). The ROS produced by PDT are well-established to destroy tumors via multifactorial mechanisms. PDT has an immediate effect on cancer cells, producing necrosis and/or apoptosis. PDT also affects the tumor vasculature, with illumination and ROS generation leading to vascular blockage, depriving the tumor of oxygen and nutrition. PDT reportedly not only kills tumor cells directly but can also induce immunogenic cell death (ICD), which causes antitumor immunity [8,9]. As the disruption of tumor immune homeostasis progresses, tumor cells exhibit sequential changes. Cancer immunotherapy has achieved significant clinical advances in advanced cancers. However, due to a highly immunosuppressive tumor microenvironment (TME) and limited tumor immunogenicity, response rates to immunotherapy in patients with different cancers are poor. Smart nanomedicine-based techniques have recently been developed that can slightly adjust the pharmacokinetics and TME of therapeutic drugs to optimize PDT and immunotherapy for better anticancer activity. In this paper, we provide new nanomedicines for PDT-assisted cancer immunotherapy, such as hypoxia-reversing nanomedicines, nanometallic organic scaffolds, and subcellular targeted nanoparticles (NPs). In addition, we describe synergistic nanotherapeutics that boost immune responses when combined with tumor-targeted immunotherapies. Finally, the challenges and future prospects in the field of PDT-assisted cancer immunotherapy are also discussed.

## 2. Breast Cancer Biology

The breast has 15–25 milk ducts that begin at the nipple, branch into smaller ducts, and finally reach the lobular unit of the terminal duct (lobule), which consists of a terminal duct and several smaller ducts (or tubules). The inner cuboidal to columnar epithelial cells and the outer myoepithelial cells delineate the ducts and tubules. The connective tissue within the lobules consists of fibroblasts on a background of collagen and acid mucin, joined by histiocytes and lymphocytes. The interlobular stroma consists of fibrous adipose tissue and is hypocellular [10]. Each of these segments consists of tiny sacs called lobules (glands). In lactating mothers, these lobules produce milk. The lobules and auricles are connected to the nipple by the milk ducts, which carry milk to the nipple. The nipple is located in the center of the areola, the dark area of skin surrounding the nipple. The lymph nodes in the breast and armpit are part of the lymphatic system, a network of nodes and ducts that drain fluid (lymph) and carry white blood cells (immune cells involved in fighting infection). The rest of the breast consists of fat and connective tissue (or fibers) [11]. Breast cancer is usually caused by a variety of factors, most of which are genetic changes that can be inherited or lifestyle or environmental factors causing mutations in a particular gene or group of genes. Some studies have reported that breast cancer gene mutations (BRCA1 and 2) can be detected in approximately 5–10% of cases, with 25% of cases occurring in women under the age of 30 [12]. Reproductive variables, such as menopause before age 12, delayed childbearing, and childbearing after age 30, also exist [12,13]. The use of exogenous hormones in the form of birth control pills or hormone replacement products, menopause, and exposure to radiation to the chest are also known susceptibility factors for breast cancer [14].

### 2.1. Stages and Grade of Breast Cancer

Patients with newly diagnosed breast cancer often present with a lump or induration in the breast or armpit, a change in the size or shape of the breast, fluid from the nipple or an inwardly directed nipple, redness or scaling of the skin or nipple, and grooves or dimpling in the breast (an orange peel-like appearance). However, in the early stages of breast cancer, no signs may be evident. Diagnostic mammography, ultrasound, and a tissue sample (called a biopsy) examined under a microscope can provide additional clinical staging. At diagnosis, the five stages of breast cancer are based on tumor size, tumor location, lymph node status, and metastasis [15]. Stage 0 is noninvasive breast cancer, including ductal carcinoma in situ (DCIS) and lobular carcinoma in situ (LCIS). Stage I is early-stage breast cancer where the tumor is smaller than 2 cm and has not spread to lymph nodes or other parts of the body. Stage II is early-stage breast cancer where the tumor is either smaller than 2 cm and has spread to one to three lymph nodes under the arm, is between 2 and 5 cm (with or without spread to the lymph nodes under the arm), or is larger than 5 cm and has not spread outside the breast. Stage III is locally advanced breast cancer where the tumor is larger than 5 cm and has spread to the lymph nodes under the arm; alternatively, the cancer is found in more than three underarm lymph nodes or has spread to lymph nodes near the breastbone or to other tissues near the breast. Stage IV is metastatic breast cancer. Moreover, the stage of breast cancer is divided into three groups: stage I and II are early invasion; stage III is locally advanced; and stage IV is metastatic cancer. Staging provides a common method to classify cancer, allowing physicians to collaborate to arrange the best treatment for a patient [16].

Recent breast cancer profiling studies have highlighted the importance of tumor biology in breast cancer behavior and, thus, the importance of histologic grade. The Nottingham histologic score (also known as histologic grading) is a method for determining the “grade” of breast cancer [17,18]. Grading is a method for determining the aggressiveness of breast cancer tumors. The Nottingham system consists of three different scores. Under a microscope, the pathologist examines breast cancer cells and assigns a score to three characteristics: 1. tube formation—how much the tumor resembles normal cell structure; 2. nuclear pleomorphism—how much the tumor cells appear to differ from normal cells; 3. mitotic activity—how quickly the cells divide or multiply. Grade I, also known as highly differentiated disease, is assigned a score of 3 to 5. Fairly differentiated grade II is assigned a total score of 6 to 7, whereas poorly differentiated disease grade III is assigned a score of 8 to 9. Grade I tumors are less aggressive. They are also more likely to have an estrogen receptor (ER+). Grade III tumors are more aggressive and more often “triple negative”, meaning they are negative for both the hormone receptor (ER and PR) and the HER2 receptor. The Nottingham score and histologic grade are used to determine whether radiation is required after surgery (lumpectomy or mastectomy) [19]. A high-grade tumor (III) is believed to have a higher risk of recurrence, and radiation treatment is thought to reduce this risk. Grade allows the radiation oncologist to determine whether the patient would benefit from additional radiation (an extra dose to a specific area at the end of radiation), whether she is eligible for accelerated partial breast irradiation (APBI), and whether the lymph nodes should be irradiated. Grade is less commonly used to determine the need for pharmacologic therapies, such as chemotherapy and hormone therapy. The only exception is for young female patients with triple-negative malignancy in whom the lymph nodes are unaffected. In these patients, high-grade tumors indicate that they should consider chemotherapy as part of their treatment.

### 2.2. Breast Cancer Therapies

The primary cancer treatments accessible to patients include surgery, chemotherapy, hormonotherapy, and radiation therapy. Breast cancer treatment choices are determined by stage (TNM), grade, hormonal status, and Ki67 status. The treatment of nonmetastatic breast cancer is primarily aimed at removing the tumor from the breast, removing regional lymph nodes, and preventing the recurrence of metastases. Local treatment of nonmetastatic breast cancer consists of surgically removing the tumor and removing or excising the axillary lymph nodes; postoperative radiation therapy may be considered. Systemic treatment may be given before surgery (neoadjuvant), after surgery (adjuvant), or both. The breast cancer subtype determines standard systemic therapy, which includes hormonotherapy for the majority of hormone-receptor-positive (HR+) breast cancer (with some patients also requiring chemotherapy), trastuzumab-based ERBB2 antibody therapy plus chemotherapy for all ERBB2+ breast cancer (with additional endocrine therapy if concurrent HR-positive), and chemotherapy alone for triple-negative breast cancer [20]. The treatment aims for metastatic breast cancer include life extension and symptom relief. Almost all individuals with metastatic breast cancer are currently incurable. In metastatic breast cancer, the same broad categories of systemic treatment are employed as in neoadjuvant/adjuvant therapies. Only in the case of metastatic illness are local therapeutic techniques (surgery and radiation) employed for palliation.

Depending on the size of the tumor and whether it has metastasized to other organs, surgery is the preferred treatment option [21]. Surgical intervention remains the primary means of treatment for local and regional breast cancer. Surgery includes mastectomy, lumpectomy, lymph node removal, and reconstruction. Conservative surgery is contraindicated in the following cases: (1) presence of diffuse suspicious microcalcifications on breast imaging; (2) positive pathologic margins after lumpectomy; (3) disease that cannot be treated by excision of a single breast tissue region with satisfactory cosmetic results, except in a small number of patients; (4) certain collagenous vascular diseases such as scleroderma; and (5) prior radiation therapy to the affected breast [22]. The surgical procedure is undoubtedly invasive and often has negative consequences for the patient, such as inflammation, induration, tenderness, disfigured appearance of the breast, asexuality, depression, and loss of self-image [23,24]. Women who have undergone mastectomy may wish to undergo breast reconstruction, either immediately or later, to improve breast appearance after tumor surgery. The option of reconstructive surgery must be offered to all women who undergo mastectomy [25]. Mastectomy is a reasonably easy surgery that normally requires 1–2 days in the hospital. External prostheses used to treat these issues can be painful and scratchy, particularly for women with large breasts. The most serious issue after mastectomy, however, is the mental impact of physical and cosmetic damage, which can include stress, depression, and negative impacts on stature and sexual activity [26]. Breast reconstruction is frequently necessary in women with breast cancer who are unable to undergo breast-conserving treatment and have a hereditary predisposition to breast cancer. Several breast reconstruction treatments involve prosthetic implants, autologous tissue flaps, or both [27].

For breast cancer, radiation treatment may involve the entire breast or part of the breast (after lumpectomy), the chest wall (after mastectomy), and the regional lymph nodes. Whole-breast radiation after lumpectomy is a routine part of breast-conserving treatment. The past decade has seen considerable advances in the delivery of postoperative radiation that aim to optimize the treatment for each person’s anatomy and reduce acute or long-term toxicity. A meta-analysis of 10,801 patients found that radiation after lumpectomy was associated with a reduction in breast cancer recurrence (locoregional or distant) by approximately half (from 35.0% to 19.3%) and a reduction in breast cancer deaths by one-sixth (from 25.2% to 21.4%) at 10 and 15 years, respectively [28]. Moujhuri Nandi et al. reported that no local recurrence occurred, and only four patients developed metastases after hypofractionated radiotherapy; they selected 135 women, most of whom had undergone mastectomy [29]. Moreover, breast conservation surgery plus radiation treatment is also associated with very high local control rates (90–95%) in the preserved breast within 10 years of treatment; these rates are comparable to those obtained with mastectomy, with most women having a good or excellent cosmetic result.

The process of killing cancer cells with the help of certain drugs is called chemotherapy [30,31]. Depending on the patient’s condition, it can be used both before and after surgery. Chemotherapy medications include docetaxel, paclitaxel, platinum drugs (cisplatin, carboplatin), vinorelbine (Navelbine), capecitabine (Xeloda), liposomal doxorubicin (Doxil), cyclophosphamide (Cytoxan), carboplatin (Paraplatin), and others, according to the American Cancer Society [32]. However, these drugs have various side effects [33]. Metastatic or secondary breast cancer is difficult to treat but can sometimes be managed for years [34]. Chemotherapy may be prescribed to treat metastatic breast cancer to minimize or slow the progression of the disease. It may also be performed to make patients eligible for surgery. Other treatment options may be started before or at the same time as chemotherapy.

Immunotherapy improves survival in other solid tumors and is a possible treatment option for breast cancer [35]. Immune checkpoint inhibitors (ICIs), which target immunosuppressive receptors such as CTLA-4 and PD-1 to improve the cytotoxicity and proliferative potential of tumor-infiltrating lymphocytes (TILs), are among the most effective immunotherapeutics [36]. ICIs, such as monoclonal antibodies targeting PD-1 (pembrolizumab, nivolumab), PD-L1 (atezolizumab, durvalumab, avelumab), and CTLA-4 (ipilimumab), have resulted in long-term responses in numerous tumor types (Table 1) [37,38,39]. 

## 3. The Immune System in Breast Cancer

### 3.1. Role of the Immune System in Normal Breast Development

In the different stages of mammary gland development, there exist diverse immune cell populations within the mammary gland stroma. Thus, it is suggested that the immune system plays an important role in normal mammary gland growth and maturation. Moreover, Plaks et al. reported that CD11c+-antigen-presenting cells and CD4+ T helper 1 (Th1) cells can interact with each other in mammary gland cells and that this interaction contributes to epithelial remodeling, mammary gland organ formation, and ductal differentiation [54]. Inhibiting CD4+ T cell responses mediated by MHC-II abolished negative regulation and accelerated alveolar branching, indicating that CD4+ immunological responses play a protective function during normal mammary gland development. However, inhibiting MHC-I-mediated CD8+ T cell activation had no effect on mouse mammary gland growth [55]. By secreting cytokines and chemokines, macrophages and eosinophils have been demonstrated to substantially influence mammary duct dilatation during puberty [56]. Breast development continues throughout pregnancy, with regression of alveolar branching, milk secretion, and regression [57]. Macrophage invasion causes extensive apoptosis in milk-producing epithelial cells during postlactational regression, reducing milk output and residual milk content [58].

During mammary gland involution, both innate and adaptive immune responses are considered to be triggered. Gene expression analysis indicated that genes associated with neutrophil and macrophage infiltration and activation were upregulated from the first day of involution and thereafter [59]. After activation by STAT3 and NF during involution, macrophages, eosinophilic granulocytes, plasma cells, and B lymphocytes move into the lumen of the mammary duct and activate proinflammatory signals [60]. Arginase-1, a hallmark of the M2 macrophage phenotype known to be elevated during normal tissue remodeling, has been found to be expressed by macrophages in growing lobules. CD45, a general leukocyte marker, has been demonstrated to be elevated during human mammary gland lobule formation and to rise over the first 12 months of life. IHC labeling revealed an increase in CD4+, CD8+, and CD19+ cells in the growing mammary gland lobule, indicating that T and B cell immune responses are important in mammary gland involution [59]. The coordination of cell death and immunological responses is crucial during breast involution, and aberrant signaling in any of these processes may result in a tumor-friendly environment.

### 3.2. Role of the Immune System in Breast Cancer

Oncodrivers increase malignancy by promoting tumor cell proliferation and survival, making them a potential target for current breast cancer treatment. Currently known oncodrivers in breast cancer are EGFR, HER2, HER3, MET, and mucin-1 (MUC1). The levels of ERBB family receptors (EGFR, HER2, HER3, and HER4) are elevated during puberty, pregnancy, lactation, and normal breast development and play important regulatory roles. The ligand-mediated activation of EGFR is highest during puberty and maturation, stimulating breast epithelial development and ductal cell differentiation. In addition, increased HER2 expression is essential for ductal expansion and acinar cell formation during puberty and maturation. erbb3/HER3 is produced only during pregnancy, whereas ERBB4/HER4 is present during both pregnancy and lactation and is required for the formation and maintenance of goblet vesicles during lactation [60]. MET expression was found to promote ductal branching and luminal cell formation during breast maturation [61]. The activation of the HER2 pathway promotes cell proliferation and survival of HER2+ breast cancer, ultimately leading to treatment resistance, invasiveness, and metastasis [62]. HER3 has been shown to be the most potent HER2/HER3 activator of the downstream PI3K/AKT pathway, and HER3 overexpression has been associated with trastuzumab resistance, suggesting that HER3 plays an oncogenic driver role in breast cancer [63]. In TNBC without targeted therapy, the overexpression of HER3 is a prognostic factor for poor 5-year DFS and 10-year DFS OS [64,65]. The overexpression of EGFR occurs in all breast cancer subtypes but is more pronounced in invasive TNBC and IBC and is associated with tumor malignancy and poor prognosis [66,67]. Hepatocyte growth factor receptor/receptor tyrosine kinase MET (HGFR/MET, commonly known as MET) has been shown to be overexpressed in TNBC. MET is an independent risk factor for tumor recurrence, and high expression of MET is usually an important risk factor for lower 5-year survival [68]. In addition, molecular interactions between MET and ERBB receptor family signaling pathways may lead to resistance of breast cancer to HER2- and EGFR-targeted therapies [69], implying that MET is promising for the development of oncodriver-targeted therapies. Since TNBC has many tumor drivers, the combination of immunotherapy and oncodriver-targeted therapy could be beneficial, but a viable treatment option is currently lacking.

Although the “classical” genetic and epigenetic aspects of tumorigenesis (activation of oncogenes, inactivation of tumor suppressor genes, abnormal cell transformation, aneuploidy, etc.) are closely associated with the development of breast cancer, the immune system likely plays an important role in tumorigenesis. Breast cancer caused by carcinogens is associated with significant impairments in tumor formation, proliferation, and the immune response. The use of dimethylbenzene(a)anthrazine (DMBA) in rats causes thymic atrophy, leading to decreased IL-2 expression and eventually impaired T cell activation [70]. In addition, aberrant cytokine expression can lead to altered cell proliferation and differentiation in malignancies. TGF-β is an anti-inflammatory cytokine released by various immune cell morphologies, and TGF-β signaling is regulated by type I and type II TGF-β receptors. Loss or mutation of TGF-β receptors is associated with increased aggressiveness and poor prognosis in breast cancer [71]. TGF-β is a cytokine released by monocytes and lymphocytes that modulates the expression of MHC-I on the surface of tumor cells, ultimately leading to NK-cell-induced cell lysis. In breast cancer, decreased IL-10 levels are associated with increased immune evasion and cell proliferation by circumventing anticancer activity [72,73]. The proinflammatory cytokine IL-6 plays an important role in the immune response to invading pathogens. However, in neoplastic diseases, IL-6 may promote tumor growth by supporting the survival of altered cells in a hostile environment [74]. Therefore, higher IL-6 levels may serve as biomarkers to distinguish advanced cancers. In addition, IL-6 has been identified as a regulator of epithelial–mesenchymal transition (EMT) in normal breast cells, resulting in cancer cells with stem-like characteristics. In this capacity, IL-6 can form a pool of highly tumorigenic cells capable of generating multiple cell types in a given tumor [75].

Immune cell infiltration and tumor microenvironment (TME) characteristics may promote oncogenic transformation in addition to protumor effects mediated by cytokine-induced inflammatory responses. In breast cancer patients, increased infiltration of CD4+ helper T lymphocytes, CD8+ cytotoxic T lymphocytes, B lymphocytes, M1 macrophages, and NK cells is associated with improved progression-free survival (PFS) and overall survival (OS) [76,77,78]. Immune cell infiltration provides cell- or cytokine-specific cytotoxicity in the immune response against tumors. Conversely, regulatory T cell infiltration promotes a tumor-friendly immune response in the tumor microenvironment by inhibiting T cell activation and inactivating effector T cells, which is associated with greater cell transformation and poorer prognosis.

M2 macrophages, also known as tumor-associated macrophages (TAMs), are abundant in the tumor microenvironment. The shift in macrophage polarization from M1 (tumor suppressor) to M2 (tumor promoter) has been shown to be a driving force in tumor development and metastasis. TAMs promote premetastatic tumor metastasis by increasing angiogenesis, tumor motility, and overall cell survival. TAMs also regulate IL-10 and TGF-β production, induce immunosuppression, and promote tumor cell proliferation in the tumor microenvironment [79,80,81]. Several studies have identified inhibition of macrophage recruitment or polarization of the macrophage phenotype (M2 to M1) as potential therapeutic targets [82,83,84].

Little is known about the B lymphocytes that infiltrate tumors. Approximately 70% of nonbreast tumor tissues test positive for infiltrating B lymphocytes. B cells are the most common infiltrating lymphocytes in nonmalignant breast lesions and ductal cell carcinoma in situ (DCIS). This infiltration into noncancerous tissues and the early stages of disease demonstrate that B cell responses play a role in the early stages of oncogenic transformation [85,86]. Certain breast cancer subtypes appear to be more sensitive to B-cell-infiltrating antitumor responses, which are linked to extended patient life. This link might be due to the surface presentation of antigens recognized by B cells, resulting in an antitumor response [87]. Although instances of B lymphocyte infiltration causing a tumor-promoting phenotype in breast cancer have not been reported, this phenomenon has been observed in other solid tumors. Ou et al. identified more B cell infiltration in neoplastic bladder tissue than in controls, and this B cell population amplified the IL-8/androgen receptor signaling pathway, leading to activation of numerous metastasis-associated matrix metalloproteinases (MMPs) (Figure 1) [88]. 

### 3.3. Immunosurveillance and Immunoediting in Breast Cancer

Since Paul Ehrlich established the notion of possible immune suppression of carcinoma more than a century ago, the role of the immune system in cancer has been a continually developing area of study [89]. Burnett and Thomas first presented the formal notion of “cancer immune surveillance” [90]: the immune system’s function in protecting against neoplastic illness and maintaining tissue homeostasis [91,92]. The absence of an appropriate animal model hampered experimental support for the theory, finally leading to the rejection of the immunosurveillance concept. Following the discovery of the role of IFN-γ in protecting mice against transplanted, chemically induced, or spontaneous fibrosarcoma tumor growth, interest in the concept of immune surveillance has been reignited, and several studies have been conducted in various mouse models with immune dysfunction [93]. Tumor development in four mouse models lacking IFN-γ and/or STAT1 function was threefold higher than that in syngeneic wild-type mice, indicating that IFN-γ and lymphocytes mediate the tumor suppressor pathway in immune surveillance [94].

Further research in mouse models will be conducted to investigate how the intact immune system influences tumor immunogenicity and growth and to better understand the immune system’s role in regulating tumor antigenicity, antigen processing, and components of the IFN-γ pathway known as immune-sculpting tumors. In addition to the role of immune landscapes in cancer development and destiny modeling beyond host protection, Schreiber et al. proposed the notion of “cancer immunoediting” [95].

### 3.4. Immune Escape Mechanisms in Breast Cancer

The strategies by which tumor cells evade recognition and elimination by the immune system can be classified into three categories [95]. These main mechanisms include (1) decreased activation of immune cells and recognition by the immune system such as the loss of tumor antigens, the absence of antigens in the tumor itself, and decreased expression of MHC class I proteins, resulting in decreased antigen presentation by T cells and stimulation of dendritic cells (DCs) with tumor cells. The next mechanism is (2) increased cytotoxic resistance due to the amplification of proto-oncogenic signals (e.g., constitutive activation of STAT3). Proto-oncogenic signals primarily include tumor drivers HER2, EGFR, and the anti-apoptotic effector BCL-2. The last mechanism is (3) tumor cells generating adaptive immune resistance by secreting immunosuppressive cytokines (TGF-, VEGF), thereby inducing activation of Tregs and MDSCs and inhibiting immunosuppressive receptors (CTLA-4, PD-1, and Tim-3). The best-defined mechanisms of immune evasion in breast cancer include the production of suppressive immunostimulatory molecules (PD-L1, CTLA4, and LAG-3), abnormal maturation of DCs, and invasion of immunosuppressive cell populations (MDSCs, Tregs, and TAMs). The presence of immunosuppressive cytokines (e.g., IL-10, TGF-, and IDO) in the TME prevents tumor cell clearance by NK cells [96,97]. After neoadjuvant chemotherapy for TNBC, there is a strong correlation between Ras-MAPK, PD-L1, and TILs, which can be detected in the remaining tumor cells, with higher Ras/MAPK activation and lower numbers of tumor-invading lymphocytes (TILs) [98]. TILs and tumor immunogenicity have been proposed as markers of therapeutic success in breast cancer. Low tumor immunogenicity in breast cancer, on the other hand, leads to the maintenance of immunosuppression in the TME.

### 3.5. Challenges of Breast Cancer Immunotherapy

Combining immunotherapy with existing conventional therapies for breast cancer, such as chemotherapy, radiation therapy, and targeted therapy, may be a beneficial treatment. Conventional cancer therapy has been proven to trigger tumor cell death, which increases antigen absorption and presentation by DCs and increases TIL recruitment. Thus, the nonoverlapping modes of action of standard medicines can turn “cool” breast cancers into immunogenic “hot” tumors [99], followed by immunotherapy to activate immunity and inhibit immunosuppression [100]. Despite the efficacy of immunotherapy in a wide range of breast cancers, only a small percentage of patients with otherwise incurable malignancies obtain life-changing long-term survival with these treatments. These findings most likely reflect the immune system’s complex and highly controlled structure. A sequence of biological procedures must be completed sequentially before efficient immune elimination of cancer cells is achievable, similar to other difficult and well-designed systems. Furthermore, the system is equipped with a number of protections, negative feedback loops, and checkpoints that allow for precise management as well as the capacity to halt and shut down an immune response. Furthermore, cancer is a complex, adaptable, and diverse disease caused by a number of genetic changes that can affect normal cellular function and activity. However, the genetic changes that are crucial to the oncogenic process might lead cancer cells to seem increasingly alien to the immune system, opening the door to immunotherapy.

Breast cancer manifests differently in various people, and tumors may differ within a patient due to changes in the clonality of cancer cells and/or the surrounding microenvironment. Furthermore, BC may be linked to chronic inflammatory disorders, while others may subvert and/or share an immune response as part of the development and metastatic process. The resulting interaction between evolving human immune system units and an emerging cancer can result in a variety of outcomes, including complete immunological eradication of the cancer, a chronic tug-of-war between the two, or uncontrolled cancer growth that has evaded an immune response, which can lead to immunotherapy resistance [101,102]. Through evolutionary processes, these pathways may be responsible for tumor-acquired drug resistance by selectively lowering the production of tumor-specific antigens [102,103]. In addition, tumor cells may contribute to escape from immunotherapy by altering enzyme activity and metabolism in the TME [102]. Tumor cells can adapt to hypoxia or vascular circulation by changing their energy metabolism, which is known as tumor metabolic reprogramming or the “Warburg effect” [104]. The lactate generated by tumor cells as a result of metabolic reprogramming acidifies the TME, affecting IFN-γ production, NK cell activation, and the number of MDSCs, resulting in a reduced immune response and increased tumor development [102,105]. Furthermore, tumor-induced acidosis stimulates TAM production and increases CTLA-4 expression on T cells. A similar study found the Warburg effect in TNBC, which leads to immunological escape of the tumor during spread [104].

One of the mechanisms that causes primary resistance to immunotherapy is the expression of a specific group of regulatory genes for various processes, such as mesenchymal transition, angiogenesis, and extracellular matrix remodeling, that are unresponsive to anti-PD-1 treatment, i.e., the innate anti-PD-1 resistance signature (IPRES) [101,106]. Other mechanisms that may contribute to immunotherapy resistance include alterations in the tumor antigen presentation pathway, which may inhibit tumor antigen presentation and may be caused by epigenetic changes related to the downregulation of antigen transporters and transcriptional inactivation of MHC class I genes [102,103].

Several strategies must be investigated to overcome immunotherapy resistance. One hypothesis is that by combining BC with medications targeting PD-1 and CTLA-4 antibodies, patient survival in other malignancies, including melanoma, might be enhanced. This approach “restores” the function and effectiveness of deactivated or depleted T lymphocytes. Conversely, blocking CTLA-4 increased the number of T effector cells in the tumor microenvironment of BCs. As a result, we recommend that the primary focus for overcoming radiation-induced immunotherapy resistance in breast cancer should be a mix of distinct signaling pathways.

## 4. Photodynamic Therapy (PDT) for Breast Cancer

### 4.1. Mechanisms of PDT

Photodynamic therapy (PDT) is a novel method to treat a wide range of disorders that require the elimination of abnormal cells. It has received much attention recently because of its specificity, minimally invasive nature, and selective cytotoxicity for malignant cells, which implies that normal cells are maintained during therapy compared to traditional treatments [107]. An advantage of PDT is that the photosensitizer can be administered in a variety of ways, such as intravenous injection or topical application to the skin. However, this delivery has implications for biodistribution. PDT works by using a specific wavelength of light to excite the photosensitizer (PS), which is selectively absorbed by the tumor tissue, causing a photochemical effect and stimulating the surrounding matrix, including molecular oxygen, to produce highly active reactive oxygen species, including singlet oxygen. These reactive oxygen species react with biological macromolecules in adjacent cells, such as carbohydrates, lipids, DNA, proteins, and enzymes, resulting in cytotoxicity, tumor cell death, and tumor blood vessel damage, which leads to tumor necrosis and detachment [108,109]. In addition, it is important to mention that high doses of PSs increase the risk of side effects (e.g., pain, erythema, non-scarring skin lesions, and death of non-tumor cells in the vicinity of the light-exposed area). Therefore, it is important to select an optimal PS dose at which PDT induces tumor cell damage with minimal damage to normal cells. When a photon of light is absorbed by a PS, it can take one of three paths: 1. PS is activated from the ground state to a short-lived excited singlet state, and the excited PS may then emit fluorescence back to the ground state. 2. The abovementioned short-lived stimulated singlet state of PS can undergo an intersystem crossing event to create a comparatively long-lived triplet state. 3. The excited triplet state PS can react with some endogenous chemicals to create a free radical (e.g., H_2_O_2_ and O^2−^). Alternatively, the more long-lived triplet state can create ^1^O_2_ by directly interacting with molecular oxygen. Most of the time, the ROS produced by PSs in PDT is mostly related to the latter phase [110,111].

Early preparations of photosensitizers for PDT were based on a complex mixture of porphyrins called hematoporphyrin derivatives. Extensive chemical and biological research has been carried out over the past 20 years to identify new photosensitizers that belong to different classes of compounds, including porphyrins, chlorins, phthalocyanines, texafrins, and phenothiaziniums [112]. Methylene blue (MB), first extracted by the German chemist Heinrich Caro, has been recognized not only as a dye but also as a medicine that has been used in the treatment of malaria [113]. MB readily penetrates the cell membrane due to the ability of its benzene ring to concentrate in the mitochondria, lysosomes, and double-stranded DNA. Because of the phenothiazinium chromophore of MB, it absorbs light near 630–680 nm, resulting in the formation of reactive oxygen species (ROS), including singlet oxygen [107,114]. Hence, MB is a photosensitizer essential for PDT. In a recent study, Jesus et al. reported that MB can generate cytotoxic reactive oxygen species (ROS) from molecular oxygen and achieve specific cancer cell death or tumor tissue damage.

### 4.2. PDT-Mediated Cell Death Mechanisms

PDT mediates tumor destruction through three main mechanisms, including direct tumor cell killing, vascular damage, and immune response. Therefore, tumor localization of the photosensitizer is an important factor determining the efficacy of PDT. In recent years, a number of more selective photosensitizers have been developed. For example, MV6401 has been shown to localize selectively in tumor vessels [115]. Drug localization is well known to be determined by vascular permeability and interstitial diffusion, which depend on the molecular size, configuration, charge, and hydrophilic or lipophilic properties of the compound, as well as the physiological properties of blood vessels. The binding of the drug to various components of the tissue may also affect transport and retention in tumors.

PDT can cause three primary types of cell death: autophagy, apoptosis, and necrosis-induced cell death. Autophagy, a lysosomal mechanism that degrades and recycles intracellular proteins and organelles, can be activated by a variety of stress signals, including oxidative stress [116,117]. This mechanism, which includes ROS as one of the key pollutants, may have both cytoprotective and death-promoting effects after cancer treatment [117]. Recent research has discovered autophagy as a method for preserving cell viability following photodynamic damage [116]. Photodamage to the PS in the lysosomal compartment may impair autophagic process completion, resulting in inadequate removal of autophagic cargo. ROS-damaged cytoplasmic components may increase phototoxicity in apoptosis-competent cells.

Apoptosis is characterized by chromatin condensation, the breakage of chromosomal DNA into internucleosomal fragments, cell shrinkage, membrane vesicles, and the production of apoptotic bodies without plasma membrane rupture. Furthermore, in cells reacting to PDT, apoptosis is the most common type of cell death. After photodynamic damage, Bcl-2 family members govern mitochondrial outer membrane permeabilization (MOMP), which is assumed to be mostly independent of p53. Photodamage to membrane-bound Bcl-2 in mitochondria-associated PS may be a favorable signal for MOMP and subsequent release of caspase activators such as cytochrome c and Smac/DIABLO or other proapoptotic molecules, such as apoptosis-inducing factor (AIF). Cleavage of Bid and MOMP is induced by lysosomal membrane rupture and the release of cathepsins from photo-oxidized lysosomes.

Phototoxicity is not only induced by caspases but can also be caused by other proteases, such as caspases and nonapoptotic pathways. The inhibition of caspase protein or gene expression often only delays phototoxicity or converts cell death to cell necrosis. Recent studies have suggested that some types of necrosis may be mediated by specific signaling pathways [109,110]. Although the molecular mechanisms by which phototoxicity mediates cell necrosis are not known, several events, such as receptor-interacting protein 1 (RIP1) kinase activation, mitochondrial ROS overproduction, lysosomal damage, and intracellular Ca^2+^ overload, are known to play roles. Severe photodestruction of the inner mitochondrial membrane or intracellular Ca^2+^ overload increase mitochondrial permeability, which may promote cell necrosis and apoptosis triggered by phototoxicity. Therefore, therapeutic PDT techniques need to be developed and improved to better understand the interactions between PDT and autophagy, apoptosis, and necrosis and how these processes can contribute to better therapeutic outcomes for tumor patients.

### 4.3. PDT in Current Breast Cancer Treatment

PSs activated by local laser irradiation have recently been associated with PDT and shown to selectively damage tumor tissue rather than normal organs. PDT is a less invasive alternative to surgery. In addition, several studies have shown that PDT can enhance the immune response against tumors through a variety of approaches. Several breast cancer studies have recently shown that the combination of targeted PDT and photothermal therapy (PTT) has the potential to successfully treat HER2-positive breast cancer as a new therapeutic tool. Xu et al. found that the uptake of anti-HER2 and anti-CD44 (Cluster of Differentiation 44) antibodies was increased in tumor cells when PS 5-aminolevulinic acid was mixed with functionalized gold nanorods using the fluorescent dye cyanine 7.5 (Cy7.5) [81]. They also observed that the combination of PDT and PTT significantly increased ROS and thermogenesis in MCF-7 breast cancer cells compared with treatment alone. They found that HER2 and CD44 receptors represented a dual target that strongly promoted the uptake of PS by tumor cells. This result suggests that the combination of PDT and PTT has potent anticancer effects in breast cancer models in vitro and in vivo [118]. Gabrielle et al. demonstrated higher binding capacity and the selective uptake of MCF-7 breast cancer cells when using Pluronic^®^ 123 (P123) micelle-loaded chrysin (HYP) photosensitizers compared with normal breast cells (MCF-10A). They also found that the HYP/P123 combination induced MCF-7 breast-cancer-cell-mediated PDT cell necrosis in mitochondria and the endoplasmic reticulum [119]. Wang and colleagues found that PDT treatment improved survival and decreased tumor size in BC mice, suggesting that PDT inhibits proliferation and metastasis [120]. Hoi et al. reported that PDT significantly inhibited breast tumor growth in an in vivo mouse cancer model [121]. Duanmu et al. found that they could safely and effectively destroy multidrug-resistant MCF-7 cells in a mouse model of chemoresistant breast cancer with PDT, targeting tumor vessels and breast cancer cells (Table 2) [122].

Tumor metastasis is considered a key factor in the high risk of death during cancer development and after treatment. During metastasis, cancer cells called circulating tumor cells (CTCs) leave the primary cancer site and enter the blood or lymphatic system. CTCs spread and accumulate in adjacent tissues and distant organs, where they become malignant and worsen tumor progression. Conventional treatments, including radiation and chemotherapy, can activate the development of cancer stem cells from CTCs and worsen metastasis [123]. In contrast, PDT has little effect on physical invasion or off-target damage. Bhuvaneswari and colleagues found tumor vascular responses to bleaching or vasoconstriction by PDT, including platelet aggregation and tumor angiogenesis [124]. Light irradiation and reactive oxygen species production during PDT may block blood vessels by exerting oxidative stress on the blood. In addition, Weng et al. reported that PDT can effectively reduce metastasis by minimizing CTCs after treatment [125]. They observed the real-time and long-term dynamics of CTCs after a single PDT treatment and after surgical resection in an animal model of breast cancer and found that CTC levels were low after PDT treatment and that primary tumor recurrence was delayed in the PDT group compared with the resection group [125].

## 5. PDT-Driven Breast Cancer Immunotherapy

Conventional breast cancer therapies include surgical resection, chemotherapeutic resection, radiation therapy, and molecular targeted therapy, all of which help to treat early-stage tumors but are ineffective in treating advanced-stage patients [126]. Cancer immunotherapy, by engaging the host immune system, can prevent cancer recurrence and prolong survival in end-stage cancer patients [127]. Many immune-based treatments, such as checkpoint blockade immunotherapy, adoptive cell therapy (ACT), and cancer vaccines, have been licensed for cancer treatment thus far. Despite the numerous benefits of immune checkpoint treatment and its application in clinical oncology, a considerable percentage of patients with breast cancer remain insensitive to immune checkpoint inhibitors due to poor tumor immunogenicity [128,129]. As a result, combination immunotherapy and other therapeutic methods are receiving more attention. PDT is less intrusive than surgery. Furthermore, multiple studies have proven that PDT boosts the immune response against tumors via a variety of methods (Figure 2) [9]. 

### 5.1. PDT-Stimulated Antitumor Immune Response

As PDT’s direct cytotoxic effects create oxidative stress in the endoplasmic reticulum and photo-oxidative damage to tumor cells, calreticulin (CRT) migrates to the cell membrane during PDT and transmits an “eat me” signal, which prompts an immune response or directly leads to tumor necrosis. Necrotic tumor cells then release intracellular proteins called damage-associated molecular patterns (DAMPs), which occur 1–4 h after PDT [130]. These DAMPs stimulate immune cell activation and migration to sites of cell damage, as well as phagocytosis of wounded cells, resulting in antigen presentation and T cell activation. PDT initiates an inflammatory response fueled by neutrophils, macrophages, and other cellular components that migrate to the treated tumor [131]. The number of neutrophils increases first, and this increase is promoted by TNF-α, a byproduct of PDT [132]. Although macrophages proliferate and move as an initial response to PDT, they also play an important role in enhancing immune-mediated effects and are sensitive to dosage changes of PDT. Low-dose PDT appears to selectively activate macrophages. In addition, an increase in the populations of myeloid cells, monocytes, macrophages, and mast cells has been observed shortly after PDT [133]. Macrophages release lysophosphatidylcholine after PDT. This protein is a substrate in T and B cell enzymatic pathways that eventually leads to the formation of macrophage-activating factor (MAF), which induces tumor-specific cytotoxic effects in activated macrophages [132]. In addition, PDT appears to enhance the phagocytic activity of macrophages and promote their involvement in the clearance of dead and dying cells at the treatment site [134]. The end result of this process is the activation of CD8+ T cells [107]. The efficacy of PDT in both innate and adaptive responses depends on T-cell-mediated anticancer activity. Presentation of antigen to T cell receptors by neutrophils, macrophages, and dendritic cells via MHC class I proteins leads to CD8+ (cytotoxic) T cell activation and tumor-specific cytotoxicity [135,136,137]. Alternatively, MHC class II antigen presentation by APCs leads to activation of CD4+ T cells (helper cells) [138]. Another difference in T cell activity is seen in CD4+ T helper 1 cells, which are responsible for the activation of cytotoxic CD8+ T cells, whereas CD4+ T helper 2 cells promote B cell proliferation and antibody class switching, which activates macrophages.

### 5.2. PDT-Induced Immunogenic Cell Death

One of the most important prerequisites for successful cancer therapy is the ability of an anticancer drug to effectively induce immunogenic cell death in tumor cells. Although numerous cellular stressors can induce immunogenic cell death in tumor cells, the specific pattern of molecular players and the nature of death depend on the treatment technique and possibly the cancer cell type [139]. Stress-induced ROS have been shown to be a prerequisite for PDT-induced ICD in tumor cells, followed by exposure to one of the major DAMPs, CRT, and the activation of the host anticancer immune system [140,141]. Therefore, it is reasonable to assume that the direct injection of PS-targeting agents into the endoplasmic reticulum is a successful technique for cancer eradication in PDT combined with breast cancer immunotherapy. For example, some studies have shown that hypericin accumulates directly in the ER, leading to substantial ROS formation and triggering severe immunological responses during PDT [142,143].

PDT can stimulate CTL-mediated antitumor immunity while also altering the immunosuppressive microenvironment in the tumor, encouraging tumor cell death. However, adaptive immune system resistance or tumor cell invasion dramatically reduces the efficiency of this immune response. Tumor cells use adaptive immune resistance or evasion to shield themselves from host immunological responses. Programmed cell death receptor 1 (PD-1) and its ligand programmed death ligand 1 are important immunological checkpoint molecules (PD-L1). Because most malignant tumor cells express PD-L1, binding to PD-1 expressed on the surface of T cells drastically reduces cytokine production as well as T cell proliferation and activity, eventually leading to immunological resistance/evasion.

### 5.3. PDT Combined with Immune Modulatory Agents

PDT-mediated immune responses that disseminate to distant areas following local therapy appear to depend on a variety of unknown characteristics; they do not occur in all individuals. However, both local and distant immunologic responses have been routinely documented in investigations combining PDT with an immunostimulatory medication. PDT coupled with immunomodulatory medications has been found in a variety of cancer animal models to produce a sustained immune response and boost effectiveness in killing tumor cells and reducing tumor growth (including breast cancer models). Combination treatment improved tumor antigen presentation, increased T cell activation, reduced Treg expression, and successfully resisted tumor rechallenge. This phenomenon was observed under some conditions, irrespective of the photosensitizer utilized.

Xia et al. recently showed that combining PDT with the immunomodulatory drug CpG oligodeoxynucleotide resulted in delayed metastatic spread, longer life, and enhanced CD8+ T cell activation [144]. Shams et al. employed a two-stage therapy approach combined with immune boosting, in which a low dosage of PDT (immunogenic) is provided followed by a high dose; antitumor effectiveness in different tumor cell lines was inconsistent [145]. This treatment prolonged survival and delayed metastatic spread. To date, these findings have yet to be translated into the clinical setting.

A unique combination treatment strategy based on two HPs and the capacity of PDT to operate directly on tumor cells and induce antitumor immunity was proposed. In two-step PDT therapy, HPPH and Photofrin were used. Following an immune-boosting low-dose PDT therapy, a tumor-controlling high-dose PDT treatment was administered. This combination PDT therapy enhanced the number of activated tumor-specific CD8+ T lymphocytes in tumor-draining lymph nodes, which was associated with a reduction in tumor spread potential (e.g., in colon26- HA and 4T1 breast carcinomas). It was also linked to better long-term tumor growth control and resistance to tumor recurrence in treated mice.

Some startling results support the use of radicicin (also known as fontanelle) in PDT treatment in conjunction with immunotherapy. Tumor formation was significantly reduced after vaccination with the radiciclovine-based PDT cell lysate TC-1 that expresses human papillomavirus E7 and the immunoadjuvant CpG oligonucleotide at both prophylactic and therapeutic doses (ODN). When PDT cell lysates were combined with ODN injections, IFN production and cytotoxic T lymphocyte (CD8+ T cell) responses were greater than when ODN or PDT was administered alone. Similar results were obtained in a rat tumor model using radiation-based PDT of TC-1 cells in combination with adenoviral injections of interleukin-12 (AdmIL-12) [131]. In this study, the combination treatment significantly increased IFN and TNF production and the expansion of CD8+ T-cell-driven CTL subpopulations, resulting in complete tumor regression in mice with 9 mm tumors. In another study, fontanin-based PDT in combination with synthetic long peptides carrying antigenic tumor epitopes was used to treat RMA cells (an aggressive T cell lymphoma cell line generated by Rauscher murine leukemia virus) in a mouse model of therapeutic immunization [132]. This strategy resulted in a significant antitumor CD8+ T cell response. These results all suggest that current cancer therapy should be based on a combination of multiple anticancer approaches, with activation of the immune system playing a key role.

Checkpoint inhibitors, such as antibodies blocking programmed cell death protein 1 (PD-1)/programmed death ligand 1 (PD-L1), are another treatment option to induce ICD in PDT patients. He et al. used nanoscale coordination polymer core–shell nanoparticles containing oxaliplatin in the core part and PS pyrophospholipid conjugates (pyrolipid) in the shell part to provide combined anti-PD-L1 therapy (NCP-pyrolipid) of treated tumor cells and showed tumor cell exposure to CRT, anticancer immunity, increased tumor cell apoptosis, and an aspirin effect [9,115]. In another study, similar effects were reported when zinc pyrophosphate (ZnP) nanoparticles were loaded with pyrolipids (ZnP-pyro) and used in combination with anti-PD-L1 therapies [146].

### 5.4. Disadvantage of PDT and Immunotherapy in Breast Cancer

PDT combined with immunotherapy has many advantages: it does not induce resistance and it is minimally invasive. Thus, it has become an effective method for treating cancer and improving clinical outcomes. In particular, nanotechnology-derived PSs or photothermal converters can significantly improve patient survival by combining phototherapy and immunotherapy. However, several variables severely limit the efficacy of PDTs and reduce their potential to elicit immunologic responses. First, tumor hypoxia can reduce the efficacy of oxygen-dependent PDTs, and oxygen consumption by PDTs can exacerbate tumor hypoxia, creating a vicious cycle [147]. Rapid tumor growth leads to inadequate blood supply, and local oxygen depletion from PDTs exacerbates tumor hypoxia, which severely impairs PDT efficacy [148,149]. Therefore, alleviating hypoxia at the tumor is an important approach to improve the efficacy of PDT-assisted cancer immunotherapy. Recently, researchers have developed various biomaterials and therapeutic agents to reduce tumor hypoxia, including hemoglobin, catalase (CAT), manganese dioxide NPs, oxygen shuttle nanoperfluorinated compounds (nanoPFCs), hyaluronidase (HAase), and metformin (Met). Second, most of the PSs used for PDT are activated by short wavelengths (e.g., visible light 400–700 nm), resulting in limited penetration depth into living tissue [150]. In addition, tissue hemoglobin can strongly absorb visible light, which greatly hinders the conversion of PS into light [151,152]. Recent studies have reported that UCNs are nanoscale materials that convert low-energy light into high-energy light through an anti-Stokes emission process with sequential excitation of multiple photons. Compared with downconverted NPs, UCNs can absorb near-infrared (NIR) light and have a relatively high penetration depth into tissues, while the light can be converted into strong UV or visible light [153,154]. Because of this property, UCN-based PDTs have been extensively studied for tumor therapy to improve tissue penetration depth. Third, high PS concentrations usually cause aggregation-induced quenching (ACQ), which severely weakens the optical properties of PS [155,156]. During PDT, high PS concentrations in the compact core of NPs tend to induce ACQ effects, resulting in decreased ROS production and fluorescence self-quenching [157,158]. To further enhance the immune response, Zhang et al. developed a cell-membrane-fused nMOF (FM) for photoactivated cancer immunotherapy using FM derived from DCs and tumor cells. Fourth, systemic administration of PSs can cause phototoxicity due to off-target effects and accumulation in normal tissues [159,160]. As compared to normal tissue, solid tumors display various TME characteristics, such as low pH, severe hypoxia, and elevated glutathione (GSH) levels. As a result, smart stimuli-responsive nanomedicines including TME-sensitive chemical linkers or components may be able to adjust the release of their carriers. TME-sensitive NPs have the ability to intelligently design the placement and pharmacokinetics of PSs and immunomodulators to improve tumor targeting and increase PDT-guided cancer immunotherapy without causing major side effects.

### 5.5. Challenges and Future Trends in PDT-Induced ICD

Hypoxia, which is common in the tumor microenvironment, may impair the effectiveness of PDT-based ICD induction. The hypoxic process is fueled by cancer cell growth, resulting in a major imbalance between oxygen supply and demand and severe metabolic abnormalities. Pathophysiological alterations, such as tumor blood vessel distortion due to an imbalance of pro- and antiangiogenic signals, physical compression, and lymphatic system disruption, all contribute to the development of oxygen deficits in the tumor microenvironment [122]. As PDT relies on oxygen transport to generate the deadly production of ROS, hypoxia significantly lowers the effectiveness of PDT in solid tumors. As a result, finding strategies to overcome the hypoxia-related limitations of PDT is vital. The use of medicines that boost the oxygen content in the tumor microenvironment can improve the efficacy of PDT in producing ICD, an approach termed oxygen-enhanced PDT. One approach is to develop adaptive oxygen carriers or generators, such as perfluorocarbon nanoparticles utilized in clinical artificial blood applications. Because of its high oxygen capacity, perfluorocarbon has a long ^1^O_2_ lifetime, resulting in long-lasting photodynamic effects [102].

Other procedures are linked with the creation of manganese dioxide nanoparticles (MnO_2_). The breakdown of MnO_2_ in the acidic and H_2_O_2_-rich tumor microenvironment provides sufficient oxygen and enhances ROS generation, which increases PDT effectiveness. Furthermore, Mn(I) ion reduction from Mn(V) in response to highly acidic H_2_O_2_ allows for in vivo selective MRI [104]. Interestingly, MnO_2_-encapsulated core–shell gold nanocages (AuNC@MnO_2_) changed the hypoxic and immunosuppressive tumor microenvironment and demonstrated consistent PDT and ICD effects. The emission of DAMPs such as CRT, ATP, and HMGB1 is characterized by oxygen-enhanced PDT with such nanoparticles, followed by DC maturation and subsequent activation of effector cells such as CD8+ and CD4+ T cells and NK cells. In two different tumor models (CT26 colorectal and 4T1 breast cancer mice), this was found to trigger an anticancer immune response and successfully suppress tumor development and recurrence.

## 6. Conclusions

In recent years, PDT has become increasingly recognized as a viable method for generating ICD in experimental cancer treatment. However, most research has employed mouse models, and this method must be validated in a clinical environment. Furthermore, new insights into the interaction between PDT and oxygen-assisted treatment may open up new avenues for the creation of a novel cancer immunotherapy. PDT and ICD are difficult areas of study with numerous potentially interesting future uses in cancer therapy.

## Figures and Tables

**Figure 1 cancers-15-01532-f001:**
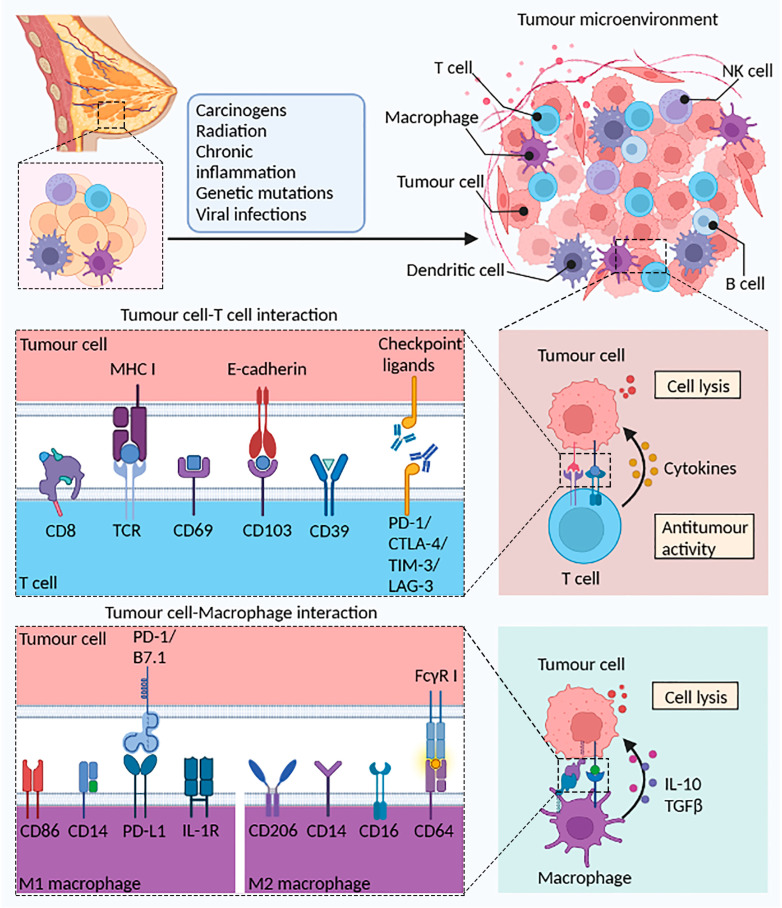
The potential role of T cells and macrophages in the TME. Normal breast tissue can become cancerous after stimulation by radiation, chronic inflammation, or viral infection. A variety of immune cells (B cells, T cells, macrophages, and DCs) elicit the body’s antitumor immune response through different mechanisms. Tumor debris generated by tumor cell death is phagocytosed and processed by DCs to present antigens to T cells, activate CD4+ Th1 cells via IFN-γ, and finally activate CD8+ CTL cells. In the tumor immune microenvironment, CD103 binds to tumor cells expressing E-cadherin, reinforcing the establishment of the link between immune cells and tumor cells. T cells highly express immune checkpoint proteins such as PD-1, CTLA-4, TIM-3, and LAG-3, which prevent tumor immune escape upon binding to the corresponding ligands on tumor cells. Additionally, increased presentation of tumor-associated antigens (TAAs) promoted the activity of M1 macrophages and upregulated the antitumor immune response. Reprinted/adapted with permission from Created with BioRender.com.

**Figure 2 cancers-15-01532-f002:**
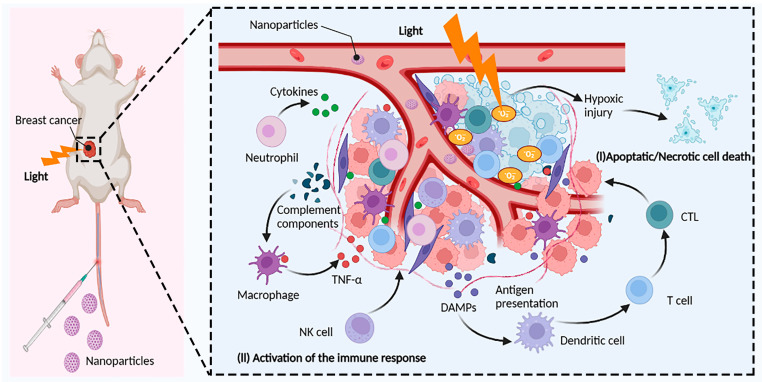
Induction of antitumor immunity by PDT. The photosensitizer (PS) is excited by light of an appropriate wavelength, and the excited PS directly transfers energy to oxygen to generate reactive oxygen species (ROS) such as singlet oxygen (^1^O_2_), superoxide anions (O_2_^−^), and hydroxyl radicals (OH) in tumor cells. Highly reactive ROS destroy tumor cells directly or indirectly through apoptotic, necrotic, and autophagy-associated cell death. In addition, PDT also induces acute inflammation and triggers the release of cytokines and stress response proteins. Initially, neutrophils are activated in the bloodstream and migrate through blood vessels to infectious or injured sites to kill cancer cells and release damage-associated molecular patterns (DAMPs). Meanwhile, blood vessel injury and tumor cells also attract macrophage infiltration, which regulates macrophage polarization and enhances macrophage phagocytosis of tumor cells. Natural killer cells (NK cells) and dendritic cells (DCs) activate adaptive immune cells such as monocytes, cytotoxic T lymphocytes (CTLs), and B cells to enhance the overall immune response by releasing cytokines. Reprinted/adapted with permission from Created with BioRender.com.

**Table 1 cancers-15-01532-t001:** Breast cancer immunotherapeutic drugs in clinical use or trial.

Category	Agent	Cancer Types	Phase	Clinical Trial Reference Number	Reference
Tumor vaccine					
Personalized peptide vaccine	MUC1 Vaccine	TNBC	Early Phase I	NCT00986609	[40]
	Folate Receptor Alpha Peptide Vaccine	TNBC	Phase II	NCT02593227	[41]
RNA vaccines	IVAC_W_bre1_uID and IVAC_M_uID	TNBC	Phase I	NCT02316457	N/A
Adoptive cell	DC-CIK cells	TNBC	Phase II	NCT02539017	[42]
	γδT cells	TNBC	Phase II	NCT02418481	[43]
HER2 vaccine	E75 peptide + GM-CSF	T1-T3 HER2 + BC	Phase III	NCT01479244	[44]
	E75 peptide (KIFGSLAFL) vaccine + GM-CSF	HER2 1+/2 + BC	Phase II	NCT01570036	[45]
	AE37 + GM-CSF	HER2 + BC	Phase II	NCT00524277	N/A
Immune checkpoint inhibitors					
PD-1	Pembrolizumab	ER+/HER2-PD-L1 + aBC	Phase Ib	NCT02054806	[46]
	Pembrolizumab + chemotherapy	High-risk, stage II/III BC	Phase II	NCT01042379	[47]
PD-L1	Atezolizumab + paclitaxel	Locally advanced inoperable TNBC/mTNBC 1st line	Phase III	NCT03125902	[48]
	Avelumab	mBC	Phase Ib	NCT01772004	[49]
	Atezolizumab + chemotherapy	TNBC	Phase III	NCT03197935	[50]
Adoptive cell therapies					
TIL Therapy	LN-145	TNBC	Phase II	NCT04111510	N/A
	Tumor infiltrating lymphocytes + IL-2	Breast Carcinoma	Phase I	NCT01462903	N/A
	CD8+ Enriched TIL vs. unselected TIL vs. unselected TIL + pembrolizumab	Metastatic BC	Phase II	NCT01174121	N/A
	Costimulated tumor-derived T cells	mBC	Phase I	NCT00301730	N/A
Dendritic cell Therapy	Neo-antigen pulsed DC	BC	Phase I	NCT04105582	N/A
	Autologous dendritic cells + chemotherapy	TNBC	Phase I/II	NCT03450044	[51]
	Celecoxib + Pembrolizumab	Brain metastases from TNBC or HER2 + BC	Phase IIa	NCT04348747	N/A
CAR-T	huMNC2-CAR44 CAR T cells	Metastatic BC	Phase I	NCT04020575	N/A
	CART-TnMUC1	TNBC	Phase I	NCT04025216	N/A
	CAR-T cells recognizing EpCAM	EpCAM + BC	Phase I	NCT02915445	N/A
	CAdVEC	HER2 + BC	Phase I	NCT03740256	[52]
Oncolytic viruses					
Oncolytic virus	Pelareorep + paclitaxel	Advanced BC/mBC	Phase II	NCT01656538	[53]
	TBio-6517 + Pembrolizumab	metastatic BC	Phase I/IIa	NCT04301011	N/A
	talimogene	ER + HER2-BC	Phase I	NCT04185311	N/A

**Table 2 cancers-15-01532-t002:** Clinical trials on PDT of breast cancer and related conditions.

Photosensitizer(s)	Wave Length (nm)	Study Details	Phase
Zinc phthalo-cyannine	675	In vitro study on murine breast cancer cell lines	Phase I
SnEt2-Purlytin	660	Clinical use for treatment of skin metastases including breast cancer	Phase I
Motexafin lutetium (Lutex)	720	Clinical use for treatment of skin metastases including breast cancer	Phase II
Photofrin	630	Clinical trial for the treatment of breast cancer skin metastases	Phase II
mono-L-aspartyl chlorin	664–667	Clinical trial for the treatment of breast cancer skin metastases	Phase II
meta-tetra (hydroxyphenyl) chlorin (m-THPC) (Foscan)	652	Patient series treatment of breast cancer metastases	Phase II
Verteporfin (Visudyne)	690	Clinical trial for treatment in primary breast cancer used in murine breast cancer models	Phase II
Porphyrins	630	Confirmed stage IIIb and IV breast cancer treatment with continuous low-irradiance PDT using verteporfin	Phase II
Chlorins	650–700	PDT study on patients with chest wall progression of breast cancer.	Phase I
Transition metal compounds	N/A	PDT for the treatment of chest wall progression of breast cancer.	N/A
Hypericin	470–570	PDT treatment of primary breast cancer diagnosed patients and patients who received mastectomy or local wide excisions of the breast.	Phase I/IIa

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
