# Peer review of "Insight into the Crosstalk between Photodynamic Therapy and Immunotherapy in Breast Cancer"

_cancers, 2023, doi:10.3390/cancers15051532_

Round 1
Reviewer 1 Report
Title: Insight into Crosstalk Between Photodynamic therapy and immunotherapy in Breast Cancer
Summary: This is a well written and informative review regarding immune modulation in breast cancer, therapy and crosstalk between photodynamic therapy and immunotherapy.
Comments:
1. Author can include disadvantage of photodynamic therapy and immunotherapy in breast cancer.
2. Author can add the Methylene blue photodynamic therapy in section 4.
3. Elaborate the role of photodynamic therapy in breast cancer metastasis by preventing circulating tumor cells.
Author Response
Reviewer 1
Comments:
- Author can include disadvantage of photodynamic therapy and immunotherapy in breast cancer.
Response: Thank you for your suggestion. We have added the disadvantage of photodynamic therapy and immunotherapy in breast cancer in revision manuscript. Please see the section of “5.4. Disadvantage of PDT and immunotherapy in breast cancer”. We have marked it in red in corresponding place.
- Author can add the Methylene blue photodynamic therapy in section
Response: Thank you for your comments. We have added the Methylene blue photodynamic therapy in revision manuscript. Please see the section of “4.1. Mechanisms of PDT”. Editorial revisions are marked in red in our manuscript.
- Elaborate the role of photodynamic therapy in breast cancer metastasis by preventing circulating tumor cells.
Response: Thank you for your advice. We have added a section to elaborate the role of photodynamic therapy in breast cancer metastasis by preventing circulating tumor cells in revision manuscript. Please see the section of “4.3 PDT in current breast cancer treatment”. We have marked it in red in corresponding place.

Reviewer 2 Report
This review about photodynamic therapy and immunotherapy in breast cancer must be extensively revised before resubmitting.
here are some major issues:
1. general information in the introduction about molecular classification is not adéquat: there are 4 and not 3 types based on molecular and histo logic findings: references for this aspect must be the whO classification 2019
2.ref 2 and ref 3 are not the adequate references for molecular classification
3. All breast cancer in situ or invasive derive from terminal tuctulo-lobular units
4.authors must review the correct anatomy and histology of the breast.
5 authors use non scientific names as: armpits instead of axial and milk ducts instead of lactiferous ducts, and molecular makeup!!!
6 did photodynamic therapy had any lethal effects on normal breast cell? And if not , explain why?
7 Is PDT a local or a systemic treatment?
8: authors confuse the concept of staging and classifying breast cancer: the stage of breast cancer is divided into 3 groups : stage I and II: early invasion; stageIII: locally advanced and stage IV metastatic.: use WHO as a reference for this section
9:the grade was not well explained: use the Nottingham grading system revised and use the appropriate reference
10: lines 110-113: this sentence must be reformulated
11: line 116 117: this sentence is not true since breast cancer treatment chooses is based on stage (TNM), grade and hormonal status and ki67 status
12 line 128: breast conservative therapy is called partial mastectomy
13: line 146: radiation treatment is a complementary treatment used after mastectomy
14: authors omit hormonotherapy as an option of treatment for majority of breast cancer HR+
Author Response
Reviewer 2
Comments:
- general information in the introduction about molecular classification is not adéquat: there are 4 and not 3 types based on molecular and histologic findings: references for this aspect must be the WHO classification 2019
Response: Thanks very much for your comments. We apologize for the mistake of not using the latest classification and we have corrected them as “According to the WHO classification 2019, breast cancer is classified into four types based on molecular and histologic findings: luminal A-like, luminal B-like, HER2-positive, and basal-like (triple negative).” in revision manuscript . We have marked it in red in corresponding place.
- ref 2 and ref 3 are not the adequate references for molecular classification
Response: Thanks very much for your comments. We have updated the ref 2 and ref 3 in revision manuscript and marked it in red in corresponding place.
- All breast cancer in situ or invasive derive from terminal tuctulo-lobular units
Response: Thanks very much for your comments. We apologize for the mistake and we have corrected them as “All breast cancer in situ or invasive derive from terminal tuctulo-lobular units, when the tumor is present only in the ducts or lobules in situ it is called Ductal Carcinoma In-Situ (DCIS)/Lobular Carcinoma In-Situ (LCIS).” in the section of abstract in revision manuscript. Editorial revisions are marked in red in our manuscript.
4.authors must review the correct anatomy and histology of the breast.
Response: Thanks very much for your comments. We are very sorry for the negligence to correct anatomy and histology of the breast, and we have corrected them in the section of “2. Breast cancer biology” in revision manuscript and marked it in red in corresponding place.
5 authors use nonscientific names as: armpits instead of axial and milk ducts instead of lactiferous ducts, and molecular makeup!!!
Response: Thanks very much for your comments. We are very sorry for the mistake and we have corrected them in revision manuscript. Moreover, we have revised the language by a native English speaker.
6 did photodynamic therapy had any lethal effects on normal breast cell? And if not, explain why?
Response: Thanks very much for your comments. Actually, PDT has very little damaging effect on normal cells. Due to the tumor cells have characteristics such as high metabolism and high oxygen consumption, and PS can be directly targeted to tumor cells rather than normal cells. In addition to the localization characteristics of PSs, it is important to mention that high doses of PSs increase the risk of side effects (eg, pain, erythema, non-scarring skin lesions and death of non-tumor cells in the vicinity of the light-exposed area). Therefore, it is important to select an optimal PS dose at which PDT induces tumor cell damage with minimal damage to normal cells. We have stated these in the section of“4.1. Mechanisms of PDT”in revision manuscript.
7 Is PDT a local or a systemic treatment?
Response: Thanks very much for your comments. According to recent studies, the anti-tumor effects of PDT derive from three interrelated mechanisms: direct cytotoxic effects on tumor cells, damage to the tumor vasculature and induction of a robust inflammatory reaction that can lead to development of systemic immunity. Thus, PDT treatment can be local or systemic, it depends to a large extent on the type and dose of photosensitizer (PS) used, time between PS administration and light exposure, total light dose and its fluence rate, tumor oxygen concentration. The PS can be administered systemically or topically. After a period of systemic PS distribution, it selectively accumulates in the tumor. Irradiation activates the PS and in the presence of molecular oxygen triggers a photochemical reaction that culminates in the production of 1O2. Irreparable damage to cellular macromolecules leads to tumor cell death via an apoptotic, necrotic or autophagic mechanism, accompanied by induction of an acute local inflammatory reaction that participates in the removal of dead cells, restoration of normal tissue homeostasis and sometimes in the development of systemic immunity. We have stated these in the section of “4.1. Mechanisms of PDT” in revision manuscript.
8: authors confuse the concept of staging and classifying breast cancer: the stage of breast cancer is divided into 3 groups: stage I and II: early invasion; stage III: locally advanced and stage IV metastatic.: use WHO as a reference for this section
Response: Thanks very much for your comments. We are very sorry for this mistake, and we have corrected it in “2.1 Stages and grade of breast cancer” in revision manuscript. We have marked it in red in corresponding place.
9: the grade was not well explained: use the Nottingham grading system revised and use the appropriate reference
Response:Thank you for your suggestion. We have corrected it in “2.1 Stages and grade of breast cancer” in revision manuscript. Editorial revisions are marked in red in our manuscript.
10: lines 110-113: this sentence must be reformulated
Response: Thank you for your comments. We have rewritten the sentence in the revision manuscript and marked it in red.
11: line 116 117: this sentence is not true since breast cancer treatment chooses is based on stage (TNM), grade and hormonal status and ki67 status
Response:Thank you for your comments. We have corrected it in the revision manuscript and marked it in red.
12 line 128: breast conservative therapy is called partial mastectomy
Response:Thank you for your comments. We have corrected it in the revision manuscript. Editorial revisions are marked in red in our manuscript.
13: line 146: radiation treatment is a complementary treatment used after mastectomy
Response:Thank you for your comments. We have corrected it in the revision manuscript and marked it in red in corresponding place.
14: authors omit hormonotherapy as an option of treatment for majority of breast cancer HR+
Response:Thank you for your suggestion. We are very sorry for this negligence and we have added the hormonotherapy for breast cancer HR+ in the part of “2.2. Breast cancer therapies” in revision manuscript. We have marked it in red in corresponding place.

Round 2
Reviewer 2 Report
All comments have been responded